



# The impact of Sahara dust aerosols on the three-dimensional structure of precipitation systems of different sizes in spring

Jing Xi[1], Yu Wang[1,2,3], Rui Li[1], Banghai Wu[1], Xiaoye Fan[1], Xinbin Ma[1], and Zixiang Meng[1]

[1]School of Earth and Space Sciences, CMA-USTC Laboratory of Fengyun Remote Sensing, University of Science and Technology of China, Hefei, 230026, China
[2]China Meteorological Administration Xiong'an Atmospheric Boundary Layer Key Laboratory, Xiong'an New Area, Baoding, 071800, China
[3]Chinese Academy of Science (CAS) Center for Excellence in Comparative Planetory, Hefei, 230026, China

*Correspondence to*: Yu Wang (wangyu09@ustc.edu.cn)

**Abstract.** Saharan dust aerosols interacting with clouds and precipitation in the Atlantic Ocean's intertropical convergence zone can significantly impact storm microphysical and thermodynamic processes. Previous satellite research often focused on individual, km-scale rain pixels, neglecting interconnections among different locations. This study innovatively employs a clustering method to group satellite precipitation radar-observed profiles into organized precipitation systems (PSs) of varying horizontal dimensions. Key features such as the mean storm top height, 85-GHz polarization-corrected microwave brightness temperature, and horizontal area with specific radar reflectivity per layer are analyzed to uncover system-level precipitation characteristics. Observations indicate that dust-laden PSs have higher storm tops, broader upper-level precipitation areas with more large particles, stronger ice scattering signals, and heavier surface rain rates than clean systems. These PSs also exhibit greater convective available potential energy (CAPE) and distinct differences in related dynamic and moisture conditions. Partial correlation and sensitivity analyses revealed that CAPE-induced changes are the primary confounding factor for dust aerosol effects. Notably, even after constraining CAPE and other thermodynamic factors, significant dust-related PS changes persist. This implies that, under comparable thermodynamic conditions, Saharan dust aerosols may enhance mid- and upper-level ice heterogeneous nucleation, thereby increasing the number of ice particles, releasing extra latent heat, and invigorating storms. Overall, this study offers a novel perspective on how dust aerosols affect organized precipitation systems.

## 1 Introduction

Mineral dust is a key component of the Earth system with important impacts on weather and climate dynamics, the Earth's radiative budget, cloud microphysics, and atmospheric chemistry (Knippertz and Stuut, 2014; Ryder et al., 2019). Dust particles coated with soluble materials (e.g., sea salt and anthropogenic pollutants) can serve as effective cloud condensation nuclei (CCN), decreasing the size of the cloud droplet for a given liquid water content, which makes collision and coalescence processes less efficient, and delays or suppresses warm rain onset and development (Kaufman et al., 2005;



Rosenfeld et al., 2001). However, there are also studies that have reached the opposite conclusion that dust aerosols can act as giant CCN and enhance precipitation (Kelly et al., 2007; Van Den Heever et al., 2006). Furthermore, mineral dust is found to be the most important natural source of ice nuclei (IN) in mixed-phase clouds and cirrus clouds (Cziczo et al., 2013; Demott et al., 2003; Sassen et al., 2003; Tobo et al., 2019). A series of satellite-based studies have shown that, compared

with homogeneous nucleation, the dust IN effect can enhance the heterogeneous freezing process and initialize ice nucleation at warmer temperatures and lower super saturations (Li et al., 2017; Min and Li, 2010; Patel et al., 2019). This effect can also influence the vertical structure of precipitation (Dong et al., 2018; Li and Min, 2010; Min et al., 2009). For instance, Zhu et al. (2023) found that precipitating drops under dusty condition grow faster in the middle atmospheric layers (with a temperature of between -5 and +2 °C) but slower in the upper and lower layers compared with their pristine counterparts.

Note that previous satellite-based observational studies on the dust effects on precipitation have mainly taken an isolated raining pixel or a single rainfall profile as the basic research unit (Patel et al., 2019; Zhu et al., 2023; Jiang et al., 2018; Han et al., 2022; Zhang et al., 2020). In fact, precipitation is spatially organized with certain three-dimensional (3-D) structure (Zhang and Wang, 2023; Li et al., 2022; Houze, 2014), i.e., precipitation system (PS), which can reflect the combined effects of multiple complex physical processes occurring inside it, including dynamic, thermodynamic, and microphysical processes

(Houze, 2014; Li and Min, 2010; Zhang et al., 2022). Aerosols can modify the cloud hydrometeor profiles and phase changes, in turn altering the dynamics and thermodynamics of one PS through latent heat release (Min et al., 2009; Zhu et al., 2024). Therefore, grouping precipitation pixels with similar formation processes in advance into the same PS and studying dust effects on its 3-D structure can provide a new perspective for better understanding the interactions between aerosols and precipitation.

Several typical individual PS cases have been used to investigate dust aerosol effects. Based on multi-platform observations, Min et al. (2009) and Li and Min (2010) investigated the impact of dust aerosols on a well-developed mesoscale convective system (MCS) occurred in the tropical eastern Atlantic and found that more small-sized cloud particles and precipitating hydrometeors formed in the dust-laden cloud system. The dust-induced microphysical effect is more evident for stratiform precipitation, where the rain rate at higher altitudes is enhanced. The simulation of the same MCS case (Gibbons et al., 2018)

indicated that increasing the IN concentration in dust scenarios results in the formation of more numerous small ice particles in the heterogeneous nucleation regime (between -38 and -5 °C) than in the clean case. Depositional growth and riming of these particles release more latent heat, which in turn invigorates convective updrafts. However, the particle growth rates decrease due to the increased competition between individual particles for limited water vapor during diffusional growth, as well as available small drops or crystals during collection processes. Chen et al. (2019) investigated the effects of dust

particles acting as CCN and IN on the growth of hail in convective clouds through a storm case simulation. The results showed that increasing CCN concentration led to larger supercooled liquid water content and larger size of hail particles, while more IN caused a decrease in the size of graupel and suppressed the growth of hailstones. Although these case studies have demonstrated the possible impacts of dust aerosols on PS to some extent, it is still unclear how the 3-D structure of PS responds to changes in dust loading, especially from a statistical perspective.





It should also be noted that the case studies mentioned above have mainly focused on MCSs with high organization and large areal coverage (Li et al., 2017; Gibbons et al., 2018; Huang et al., 2019), while not enough attention has been paid to smaller PSs. Based on CloudSat radar observations, Douglas and L'ecuyer (2021) found that aerosols can increase the precipitation formation rate and in-cloud vertical motion in warm clouds, and this effect is more pronounced when the warm rain system is smaller in size. In fact, aerosols and PSs are simultaneously influenced by the meteorology conditions, which compose a

buffered system with complex feedback processes (Stevens and Feingold, 2009; Rosenfeld et al., 2014; Guo et al., 2018). Numerous studies have shown that meteorological factors, such as convective available potential energy (CAPE), relative humidity (RH), wind shear, and other parameters, can modify the effects of aerosols on clouds and precipitation (Zhao et al., 2024; Zhu et al., 2024; Chen et al., 2020; Zang et al., 2023). In general, the larger horizontal coverage and more organization the PS has, the stronger instability energy and more abundant water vapor supply can be found (Fu et al., 2021; Zhou et al.,

2013), which may obscure or even cancel out the effect of dust on precipitation. In order to deepen the understanding of the dust aerosol effects, therefore, it is necessary to investigate the properties of PSs of different sizes as a function of dust loading and meteorology conditions.

In this study, the impact of dust aerosols on PSs is statistically investigated based on the long-term satellite observations. The tropical Atlantic Ocean (30°W-10°E, 5°S-25°N) on the western side of Africa is chosen as the study area. The large

quantities of dust aerosols are transported here by trade winds in springtime from the adjacent Sahara desert (Choobari et al., 2014; Prospero et al., 2021), and interact with the frequently occurred PSs over the Atlantic intertropical convergence zone (ITCZ) (Li and Min, 2010; Rosenfeld et al., 2014), making this region an ideal testbed for studying the interactions between dust aerosols and precipitation (Dong et al., 2018).

This paper is organized as follows: Section 2 introduces the data and methods used in this study. Section 3 presents the

variations of 3-D structure of PSs with different sizes and different dust concentrations, and explores the influence of meteorology conditions. Sections 4 and 5 include the discussion and conclusion of the main findings of this study.

## 2 Data and methods

### 2.1 Dataset

The Tropical Rainfall Measuring Mission (TRMM) satellite is a joint mission between the National Aeronautics and Space

Administration (NASA) of the United States and the National Space Development Agency (NASDA) of Japan, and is designed to improve our understanding of the distribution and variability of tropical and subtropical (between 38°N and 38°S) precipitation (Kummerow et al., 1998; Zipser et al., 2006). It carries the first space-borne precipitation radar (PR) and a nine-channel passive microwave radiometer, the TRMM Microwave Imager (TMI). The former operates at 13.8 GHz with a 215 km swath, enabling the capture of the 3-D storm structure. The latter measures brightness temperatures at the following five

frequencies: 10.65, 19.35, 21.3, 37.0, and 85.5 GHz and has a 759 km swath (Kummerow et al., 1998; Fu and Liu, 2001), providing information on a variety of geophysical parameters including liquid and solid hydrometeors in the atmosphere.



12 years (2003-2014) of TRMM observations from March to May are used, including PR level-2 product 2A25 and TMI level-1 product 1CTRMMTMI. The 2A25 product is widely used in global precipitation analysis (Liu and Fu, 2001; Liu et al., 2008; Nesbitt et al., 2006), providing vertical profiles of attenuation-corrected radar reflectivity factor and rain rate with a

horizontal (vertical) resolution of 5 km (250 m). Each precipitation profile is classified into three rain types including stratiform, convective and other, based on the vertical and horizontal variability in radar reflectivity (Awaka et al., 2009; Schumacher and Houze, 2003). The vertically and horizontally polarized brightness temperatures at 85.5 GHz from 1CTRMMTMI product are used to calculate the polarization corrected temperature (PCT85), as an indicator of the amount of ice scattering within the cloud (Vivekanandan et al., 1991; Wall et al., 2014). Considering the inconsistent horizontal

resolutions of PR (~4.5 km at nadir) and TMI (5 km×7 km at 85.5 GHz), their measurements are merged using the moving surface fitting method, with the PR measurements acting as the target data (Fu et al., 2013).

Similar to Xi et al. (2024), PSs are identified as contiguous precipitation area consisting of pixels with PR near-surface rain rate greater than 0 mm/h. It should be noted that only PSs untruncated by the edge of the PR swath were used in our study to ensure completeness. Additionally, PSs should contain at least four precipitating pixels (covering more than 80 km2) to

reduce uncertainties in subsequent statistical analysis.

As shown in Table 1, the selected parameters describing PS properties of 3-D structure and intensity are divided into two categories: general characteristics and vertical profiles. The former includes the precipitation area, mean storm top height, mean surface rain rate, maximum 30 dBZ echo top height, convective precipitation percentage and area fraction of PCT85 ≤ 250 K. In these characteristics, maximum 30 dBZ echo top height represents the maximum altitude that large hydrometeors

can reach, which is often used as an indicator of convective intensity because intense updrafts are required to lift large hydrometeors to higher altitudes (Liu and Zipser, 2013, 2015). In addition, the area fraction of PCT85 ≤ 250 K represents the fraction of ice-phase hydrometeors in the PS (Toracinta et al., 2002). The vertical characteristics of PS are composed of variables such as the median rain rate, maximum radar reflectivity and precipitation area of 20, 30 and 40 dBZ. It should be mentioned that the mean rain rate is typically used to represent precipitation intensity when performing statistical analysis on

isolated raining pixels. Within a complete PS, however, the distribution of precipitation rates tends to follow a gamma distribution (Martinez-Villalobos and Neelin, 2019; Fan et al., 2024), with a majority of weak precipitation largely determining the average value. Therefore, the median rain rate is used here to indicate the precipitation intensity of the entire PS.

Aerosol data are obtained from Moderate Resolution Imaging Spectroradiometer (MODIS) sensors onboard Terra and Aqua

satellites, which are in sun-synchronous orbits crossing the equator at 10:30 and 13:30 local time, respectively (Barnes et al., 2002). Level-3 collection 6 daily aerosol optical depth (AOD) products (MOD08_D3/MYD08_D3) with a 1°×1° grid are used (Levy et al., 2013; Remer et al., 2005).






**Table 1. Definitions of parameters describing the PS 3-D structure.**

| | Properties | Descriptions |
|---|---|---|
| General Characteristics | Area (km²) | Horizontal size of PS, is calculated as the number of raining pixels multiplied by the size of each pixel (~20.35 km²). |
| | Mean Storm Top Height (km) | Vertical height of PS, is calculated as the average storm top height of all pixels. |
| | Mean Surface Rain Rate (mm/h) | Precipitation intensity of PS, is calculated as the average surface rain rate of all pixels. |
| | Maximum 30 dBZ Echo Top Height (km) | Convective intensity of PS, is defined as maximum 30 dBZ echo top height of all pixels. |
| | Convective Precipitation Percentage (%) | Fraction of convective precipitation of PS, is defined as number of convective precipitation pixels to the number of total pixels. |
| | Area Fraction of PCT85 ≤ 250 K (%) | Fraction of ice-phase hydrometeors of PS, is defined as number of pixels that PCT85 ≤ 250 K to the number of total pixels. |
| Vertical Profiles | Median Rain Rate (mm/h) | Median rain rate at different altitudes of PS, represents vertical distribution of precipitation intensity. |
| | Maximum Radar Reflectivity (dBZ) | Maximum radar reflectivity at different altitudes of PS, represents vertical distribution of convective intensity. |
| | Area of 20, 30, and 40 dBZ (km²) | 20, 30, and 40 dBZ area (number of pixels with radar reflectivity greater than or equal to 20, 30 or 40 dBZ multiplied by pixel size) at different altitudes of PS, represent the area of precipitation size particles, large supercooled liquid or ice particles, and large ice particles, respectively (Liu and Zipser, 2015; Liu and Liu, 2018; Xu et al., 2009). |





Note that MODIS cannot detect aerosols below clouds (Shin et al., 2019; Christopher and Gupta, 2012), so it is difficult to

determine the dust concentration of each PS. Given that Saharan dust outbreaks can last for several days and affect an

extensive area (Knippertz and Stuut, 2014), we estimated the dust concentration of PSs using spatiotemporal interpolation

and extrapolation methods. Specifically, the available AOD from a total of 7 days (the day of the PS event, and 3 days before

and after), and from all grids extending 3˚ outward from the selected PS, were averaged to determine the dust aerosol

content corresponding to this PS. In fact, we also conducted sensitivity tests by selecting different numbers of days and grids

extending outward, and found that these changes had little impact on the classification of dust loading for the specific PS

mentioned later, with more than 75% of the classification results remaining consistent (see Supplementary Table S1).

As an example, Figure 1 shows the spatial distributions of near-surface rain rate and AOD for the 6 PSs that occurred over

Atlantic Ocean on 27 February 2008, and their general characteristics are given in Table 2. It can be seen that different PSs

are highly variable in terms of the horizontal scale, as well as other properties and dust conditions. For example, PS3 and

PS6 have similar horizontal sizes but exhibit noticeable differences in storm top heights and surface rain rates, implying

differences in thermodynamic and dynamic conditions, or possibly aerosol effects. In particular, the higher maximum 30

dBZ echo top height and greater convective precipitation percentage in PS6 demonstrate stronger convective intensity, with

a greater number of ice-phase particles. Moreover, it is interesting that PS1 and PS5 have different precipitation areas but

similar other properties, such as maximum 30 dBZ echo top height, surface rain rate and convective precipitation percentage,

which may be due to their locations in adjacent regions, resulting in relatively consistent background fields and dust aerosol

concentrations.

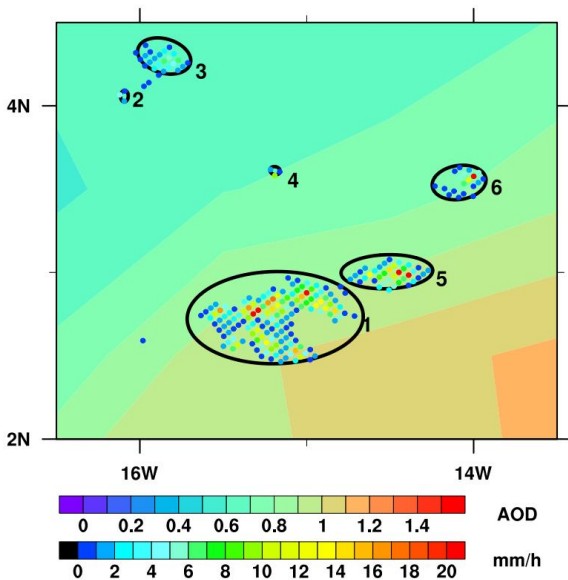

**Figure 1. Near-surface rain rates of several PSs idealized as ellipses over the tropical Atlantic Ocean on 27 February
2008 measured by PR (orbit number: 58594). The shaded map is the spatial distribution of MODIS AOD. The six PSs**
**are numbered and represented by black elliptical outlines.**



**Table 2. The general characteristics of the 6 PSs.**

|  | PS1 | PS2 | PS3 | PS4 | PS5 | PS6 |
|---|---|---|---|---|---|---|
| Mean AOD | 0.73 | 0.74 | 0.75 | 0.73 | 0.75 | 0.77 |
| Area (km²) | 2849.00 | 61.05 | 529.10 | 61.05 | 752.95 | 427.35 |
| Mean Storm Top Height (km) | 9.08 | 3.75 | 6.38 | 3.92 | 9.20 | 8.13 |
| Mean Surface Rain Rate (mm/h) | 4.74 | 2.03 | 1.88 | 4.11 | 4.57 | 2.86 |
| Maximum 30 dBZ Echo Top Height (km) | 11.25 | 3.50 | 4.50 | 1.50 | 11.25 | 7.75 |
| Convective Precipitation Percentage (%) | 58.57 | 100.00 | 38.46 | 66.67 | 54.05 | 42.86 |
| Area Fraction of PCT85 ≤ 250 K (%) | 65.00 | 0.00 | 0.00 | 0.00 | 43.24 | 19.05 |



In addition, a set of meteorological variables (Table 3) are used to analyze the influence of meteorology conditions on aerosols affecting cloud and precipitation, which are obtained from the fifth generation European Centre for Medium-Range Weather Forecasts (ECMWF) atmospheric reanalysis of the global climate (ERA5) with high spatial and temporal resolutions (0.25° and 1 h, respectively) (Hersbach et al., 2020). They are matched to the closest PS pixel spatiotemporally. The average of all pixels within one PS is then used to represent the corresponding meteorological field.


**Table 3. Definitions of variables describing PS meteorology conditions.**

| | Properties | Descriptions |
|---|---|---|
| Original Variables | CAPE (J/kg) | Convective available potential energy |
| | VV500 (m/s) | Vertical velocities at 500 hPa |
| | VV850 (m/s) | Vertical velocities at 850 hPa |
| Calculated Variables* | 10mWS (m/s) | Wind speed at 10 m |
| | VWS ($10^{-3}s^{-1}$) | Vertical wind shear (VWS) between 850 hPa (~1.5 km) and 500 hPa (~5.5 km) |
| | LTS (K) | Lower tropospheric stability, calculated as the difference between potential temperature at 700 hPa and 1000 hPa |
| | RHlow (%) | Water vapor situation at low-level, calculated as the average of relative humidities from 1000 to 850 hPa (Chen et al., 2017) |
| | RHmid (%) | Water vapor situation at mid-level, calculated as the average of relative humidities from 700 to 400 hPa (Chen et al., 2017) |

*: The details of the calculation methods are given in Supplementary Text S1.

## 2.2 Classification of PSs

Considering that convective cloud systems driven by thermal convection are more susceptible to aerosols compared to synoptic precipitation caused by large-scale uplift associated with frontal or low-pressure systems (Guo et al., 2017; Guo et al., 2019), only those PSs that have at least one embedded convective core were used here. Moreover, since the dust IN effect is the focus of this study, the precipitation height of the selected PSs was further required to be higher than 6 km to reduce the possible confusion caused by warm rain processes.



In this study, we attempted to investigate the effects of dust on precipitation by comparing the differences in the general characteristics and 3-D structure of PSs under clean and dusty conditions (Guo et al., 2018; Koren et al., 2012), using the 30th (0.24) and 70th (0.36) percentiles of AOD for all PSs as thresholds (Supplementary Figure S1), respectively. To further analyze the relationship between dust effects and PS size, we categorized the collected PSs into three types based on precipitation area: small-sized (< 2000 km²), medium-sized (between 2000 km² and 10000 km²), and large-sized (> 10000

km²), and thus performed a comparative study.

Figure 2 shows the spatial distributions of all dust-free and dust-laden PSs of different sizes during spring from 2003 to 2014. Obviously, small-sized PSs have the largest number of samples, while large-sized ones have the smallest. Most of the PSs are located in the ITCZ, where the convergence of warm and moist trade winds brings updrafts favoring the development of convection and precipitation (Schneider et al., 2014). Dust-laden PSs are closer to land than clean ones, which is consistent

with the distribution of the multi-year average AOD (the shaded background in Fig. 2) due to the transport of dust aerosols from land. This suggests that the proposed method dividing PSs into clean and dusty conditions is reasonable. It should be noted that the closer the PSs are to the coastline, the stronger the thermodynamic conditions are likely to be. As a result, the influence of meteorology conditions needs to be carefully considered later in this study.

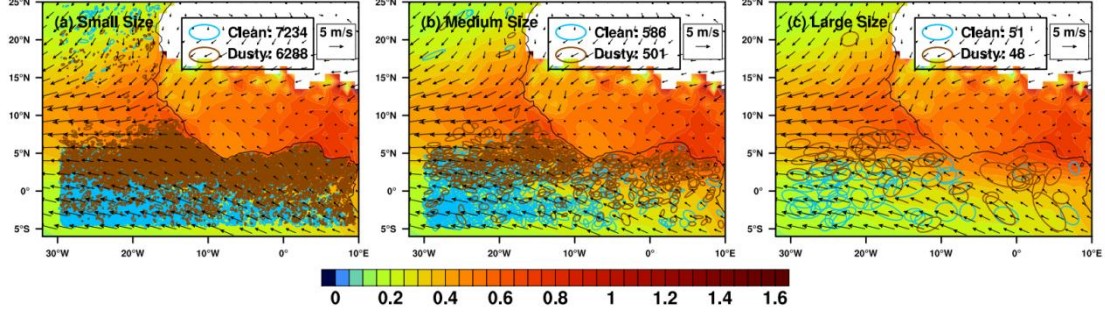


**Figure 2. Spatial distributions of (a) small-sized, (b) medium-sized and (c) large-sized of PSs idealized as ellipses. Blue and brown ellipses denote clean and dusty PSs respectively, and the numbers in the legends denote the sample numbers. The fields are composites of average AOD from MODIS (shaded) and wind vectors at 850 hPa from ERA5 (m/s) during 2003-2014 (March to May).**


## 3 Results

### 3.1 Dust effects on the 3-D structure of PSs

Figure 3 shows the mean values of the general characteristics of PSs of different sizes under clean and dusty conditions, respectively. Regardless of PS size, almost all the parameters including the mean storm top height, mean surface rain rate,




maximum 30 dBZ echo top height, convective precipitation percentage and area fraction of PCT85 ≤ 250 K are

significantly higher under dusty conditions than those under clean conditions, indicating that dust aerosols would invigorate

PSs with stronger convective activity, more ice-phase particles and more precipitation.

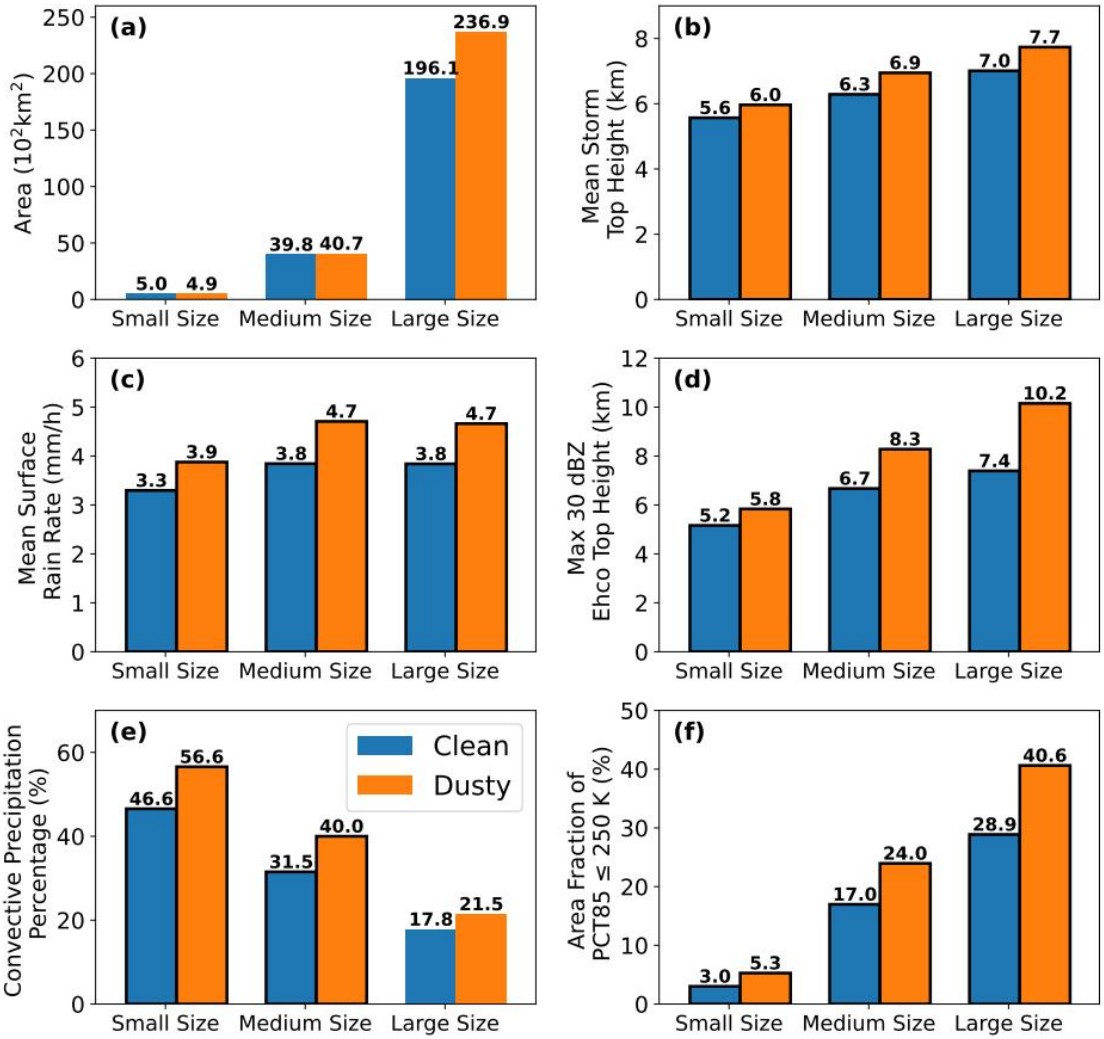

**Figure 3. The averages of general characteristics of PSs of different sizes, including (a) area (km²), (b) mean storm top height (km), (c) mean surface rain rate (mm/h), (d) maximum 30 dBZ echo top height (km), (e) convective precipitation percentage (%) and (f) area fraction of PCT85 ≤ 250 K (%) under clean and dusty conditions. The average values of these properties are labeled on the data bars. The black border around the data bar indicates that the difference between clean and dusty conditions is statistically significant at the 95% confidence level using a**
**Student's t-test.**

 

Regarding the vertical characteristics of precipitation, it can be seen that the median rain rate increases significantly at all heights with the presence of dust, regardless of PS size or precipitation type (Fig. 4). In particular, the enhancement in convective precipitation is more pronounced than that in stratiform precipitation. This suggests that dust aerosols can enhance the precipitation intensity of PSs, mainly by increasing convective precipitation. In addition, the difference in rain rates above the freezing level between dust-laden and dust-free scenarios increases with PS size, implying that the impact of dust on solid precipitation is more obvious for larger PSs.

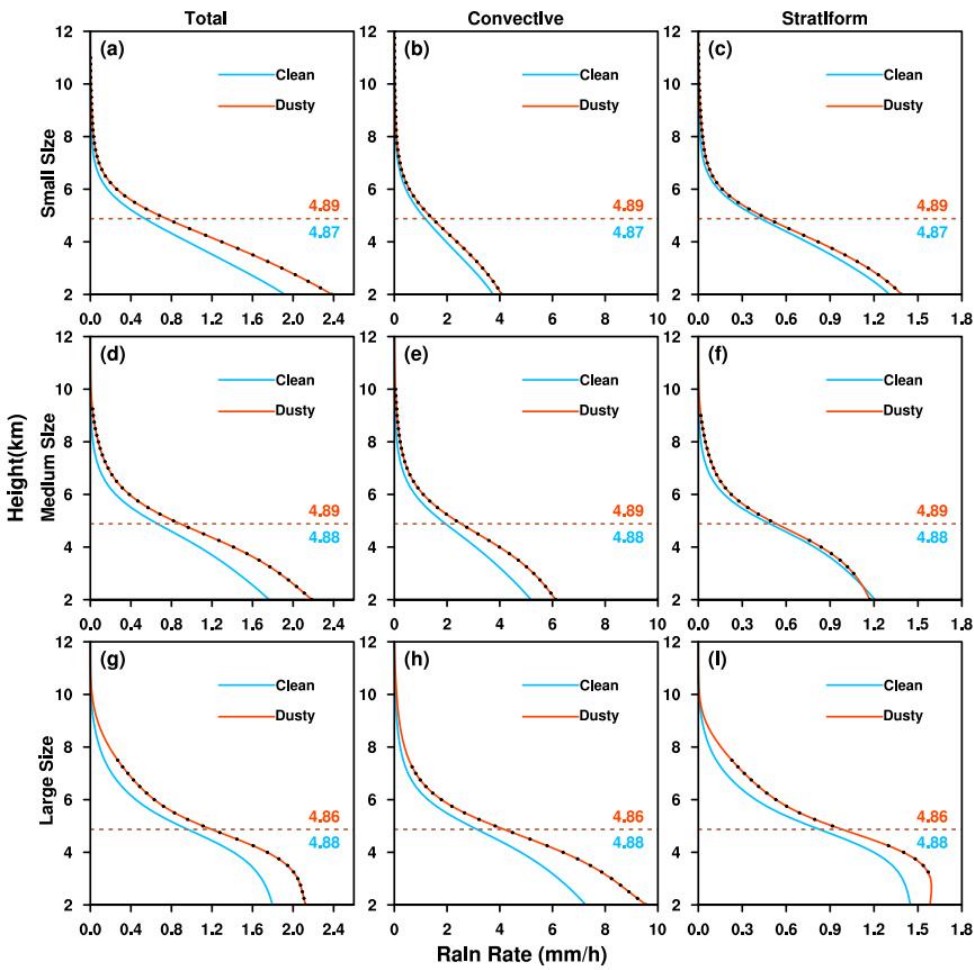

**Figure 4. The mean vertical profiles of median rain rates for total, convective and stratiform precipitation of small-sized (upper row), medium-sized (middle row), and large-sized (bottom row) PSs under clean and dusty conditions. The black dot on the red line indicates that the difference between clean and dusty conditions is statistically significant at the 95% confidence level using a Student's t-test. The horizontal dashed lines show the freezing level with values labeled.**





The average 20 dBZ, 30 dBZ and 40 dBZ area profiles for various sizes of PSs under clean and dusty conditions are shown in Figure 5, respectively. For convective precipitation, all reflectivity areas are significantly enhanced at all heights under dusty conditions, regardless of the PS size (Figs. 5b, e, and h). A similar conclusion is made for total precipitation, which indicates a substantial increase in the number of precipitation particles of different phases and sizes due to the presence of

dust. In contrast, for stratiform precipitation, this increase (mainly 20 dBZ) is only present above the freezing level (Figs. 5c, f, and i), because more ice particles due to the dust IN effect can be advected into stratiform regions with convection outflows at upper levels within a PS (Min et al., 2009; Li and Min, 2010). In those small- or medium-sized PSs, the 20 dBZ area below the freezing level is even reduced significantly under high dust loading. This decrease may be attributed to the fact that the warm and dry Saharan Air Layer (SAL) can interact with cloud and precipitation systems along its path across

the tropical Atlantic (Li and Min, 2010; Braun, 2010; Barreto et al., 2022) and enhance the evaporation process (Dong et al., 2018; Li and Min, 2010). In particular, the cloud margin under sub-saturation is more susceptible to evaporation effect caused by the entrainment of ambient dry and warm air than the cloud core (French et al., 2009; Sun et al., 2023). This makes the reduction in the precipitation area more pronounced for small-sized PSs, although the median precipitation rate still increases due to the presence of dust (Fig. 4c). As the PS size increases, dynamic and thermodynamic forcings such as

sustained large-scale water vapor transport and convergence become stronger, so that the particle size becomes larger, which in turn makes the evaporation effect less significant. For large-sized PS, therefore, the increase in various reflectivity areas at different altitudes due to dust aerosols can be observed, even in stratiform regions (Fig. 5i).

Figure 6 shows the vertical profiles of the 10th, 50th, and 90th percentiles of maximum radar reflectivities as proxies of various convective strengths (Zipser and Lutz, 1994; Xu et al., 2009). It is clearly seen that the maximum radar reflectivities

at different percentiles increase at all heights in response to higher aerosol loading, regardless of PS size or precipitation type, which again suggests that dust can stimulate stronger convective activity in PSs. In addition, the difference in the 90th percentile of maximum radar reflectivity between clean and dusty conditions is greater than that for the 10th percentile. This might be explained by the fact that stronger updrafts within a well-developed PS can lift dust particles to higher altitudes more easily, and allow more smaller ice particles produced by dust IN effect to grow to larger size. Similar enhancement in

convective precipitation with increasing dust aerosols have also been found in previous modeling and observational studies (Van Den Heever et al., 2006; Zhang et al., 2020; Zhang et al., 2021).



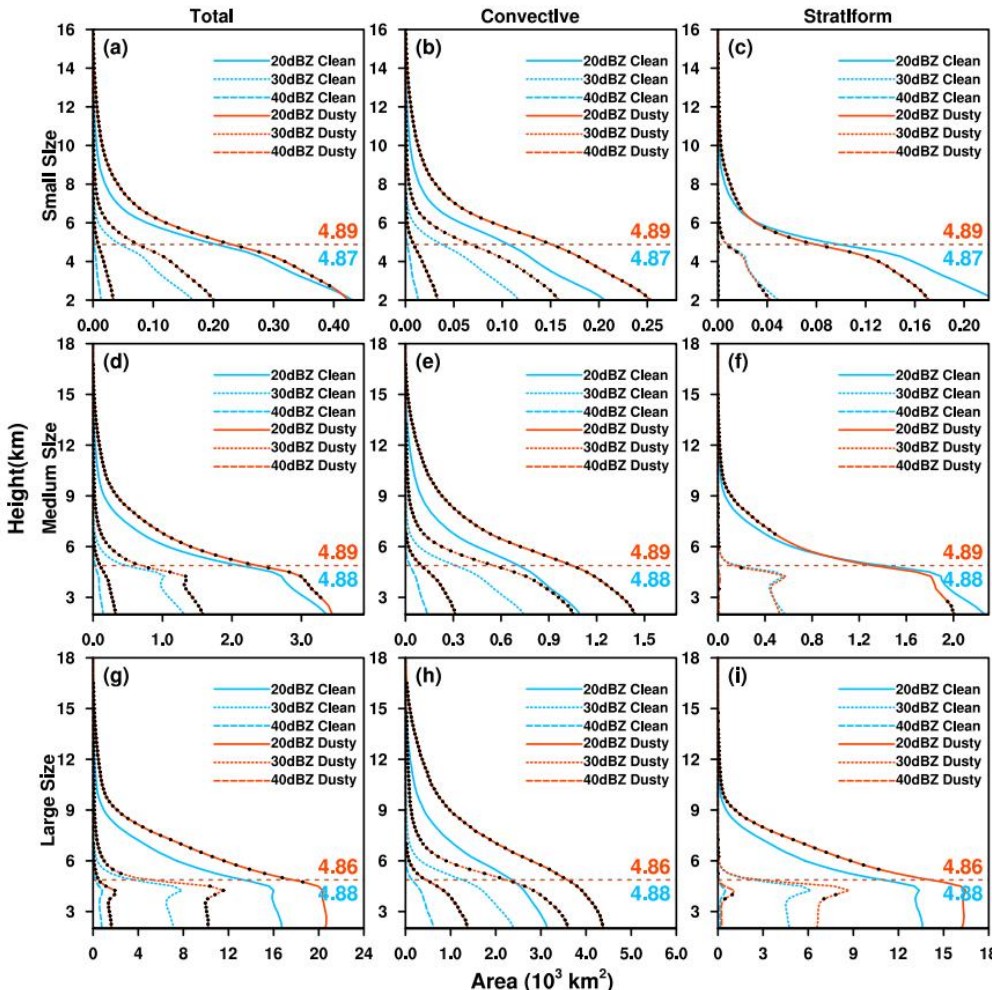

**Figure 5. The mean vertical profiles of 20 dBZ, 30 dBZ and 40 dBZ areas for total, convective and stratiform precipitation of small-sized (upper row), medium-sized (middle row), and large-sized (bottom row) PSs under clean and dusty conditions. The black dot on the red line indicates that the difference between clean and dusty conditions is statistically significant at the 95% confidence level using a Student's t-test. The horizontal dashed lines show the freezing level with values labeled.**



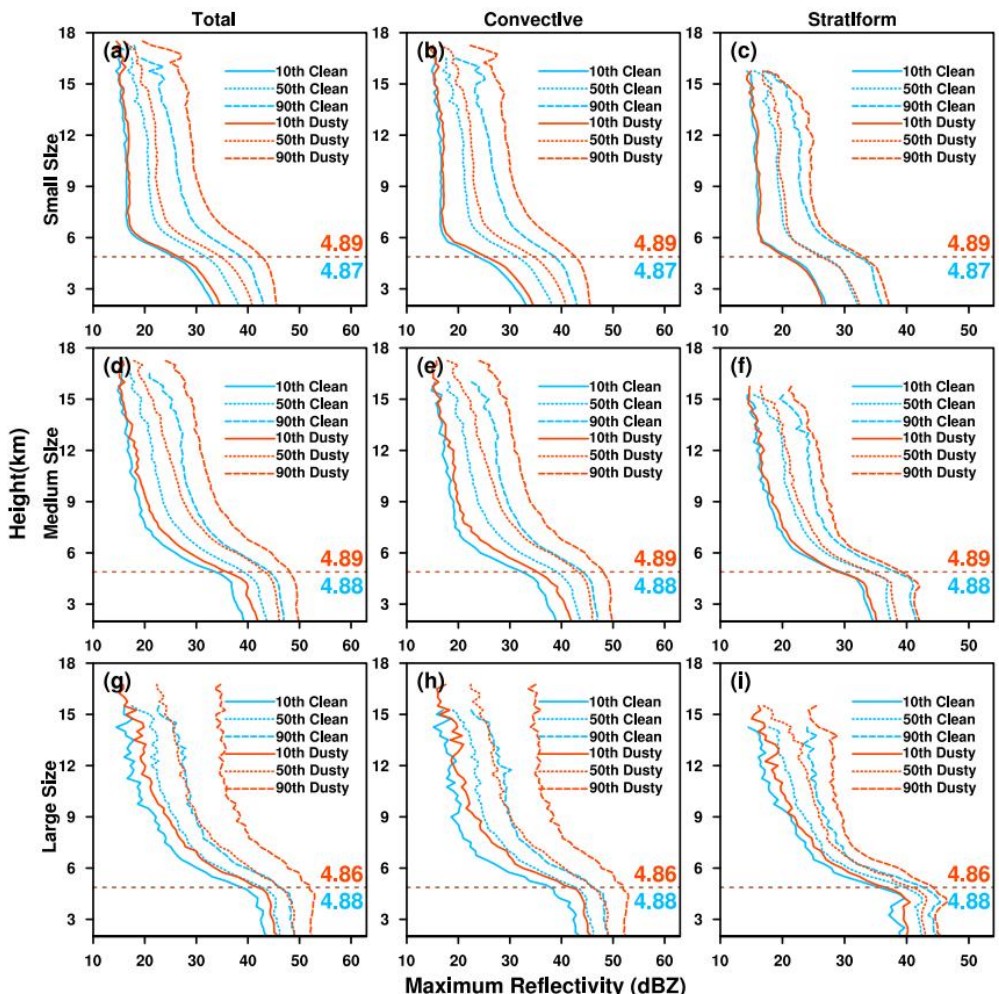

**Figure 6. Vertical profiles of the 10th, 50th, and 90th percentiles of PS maximum radar reflectivity for total, convective and stratiform precipitation of small-sized (upper row), medium-sized (middle row), and large-sized (bottom row) PSs under clean and dusty conditions. The horizontal dashed lines show the freezing level with values labeled.**

## 3.2 Influence of meteorology conditions

It has been a great challenge to disentangle aerosol effects from meteorology conditions in observational studies (Rosenfeld et al., 2014; Zhu et al., 2023). In fact, the interactions among meteorology, precipitation and aerosols are complex. As mentioned above, the warm and dry SAL associated with dust outbreaks enhances the evaporation effect and masks the dust enhancement effect below the freezing level for 20 dBZ area of stratiform precipitation (Fig. 5). In addition, the stronger



dynamic and thermodynamic conditions related to large-sized PSs could amplify the dust enhancement effect on convection, as seen in the variation of maximum radar reflectivity (Fig. 6).

On the other hand, the differences in the zonal distribution of PSs between dust-free and dust-laden scenarios (Fig. 2) may also be accompanied by variations in meteorological fields, due to the strong land-sea thermal contrast. The high surface temperatures near land and abundant moisture inflow from the ocean increase atmospheric instability in coastal areas,

resulting in higher CAPE values (Roy Bhowmik et al., 2008; Alappattu and Kunhikrishnan, 2009). In this regard, Figure S1a shows that the multi-year average CAPE is indeed higher in areas closer to the coastline. Also, it is interesting that the mean CAPE over the ocean for PSs with heavy dust is higher than that under pristine conditions (Fig. S1d). Therefore, the influence of the background field should not be ignored when studying dust effects on PSs, although in above analysis we have attempted to constrain it through classifying PSs by size.

Table 4 presents the statistical results of several meteorological parameters for clean and dusty PSs of different sizes, respectively. It is remarkable that dust-affected PSs exhibit stronger thermodynamic conditions (significantly higher CAPE and lower LTS), which are more favorable for the development of PSs. Also, the VWS increases significantly, similar to previous studies (Dunion and Velden, 2004). This may be due to the fact that PSs with dust aerosol loading are mainly located in the SAL, where the midlevel African easterly jet centered near 700 hPa can greatly increase the low-level VWS.

The intrusion of dry and warm air here (Dong et al., 2018) also causes lower RHmid in these PSs. On the other hand, the meteorological conditions favorable for large-sized PSs, such as strong atmospheric instability and persistent large-scale moisture transport and convergence (Zhou et al., 2013), are always similar regardless of dust concentration. In contrast, the differences in meteorological conditions between pristine and dusty conditions are more significant for small- and medium-sized PSs.





**Table 4. Statistical characteristics of the meteorological parameters of PSs of different sizes under clean and dusty conditions. The bold font indicates that the difference between clean and dusty conditions is statistically significant at the 95% confidence level using a Student's t-test.**

| | Small-sized | | Medium-sized | | Large-sized | |
|---|---|---|---|---|---|---|
| | Clean | Dusty | Clean | Dusty | Clean | Dusty |
| 10mWS (m/s) | **4.05±1.67** | **3.53±1.59** | **4.10±1.64** | **3.40±1.44** | 4.00±1.65 | 3.66±1.24 |
| CAPE (J/kg) | **438.43±265.26** | **682.11±397.89** | **392.70±211.76** | **562.21±318.20** | **382.85±135.11** | **506.38±278.41** |
| VWS ($10^{-3}\,s^{-1}$) | **1.24±0.80** | **1.54±0.87** | **1.31±0.77** | **1.58±0.87** | 1.32±0.67 | 1.55±0.79 |
| LTS (K) | **14.51±0.90** | **14.27±1.08** | **14.62±0.75** | **14.46±0.85** | 14.77±0.56 | 14.58±0.92 |
| RHlow (%) | **89.25±4.90** | **88.15±6.57** | 90.51±3.80 | 90.07±5.23 | 91.55±3.20 | 90.05±7.75 |
| RHmid (%) | **60.49±13.38** | **57.57±13.14** | **64.06±12.60** | **62.08±12.82** | 69.87±12.69 | 70.42±11.43 |
| VV500 (m/s) | 0.78±2.69 | 0.74±2.51 | 0.95±2.01 | 0.99±2.06 | 1.69±2.01 | 2.18±2.21 |
| VV850 (m/s) | **0.76±1.81** | **0.63±2.07** | **1.03±1.69** | **0.71±1.67** | 0.92±2.28 | 1.12±1.88 |





To further investigate the potential influence of meteorology conditions, the total correlations between AOD and PS characteristics, and the partial correlations with the effects of meteorological parameters eliminated are calculated (Figure 7), similar to the methods of Xi et al. (2024). More details of the calculation method for partial correlations are given in

Supplementary Text S2. Generally speaking, if the partial correlation for a certain meteorological factor is similar to the corresponding total correlation, it means that this parameter does not affect the correlationship between AOD and PS characteristics. In particular, if the total correlation coefficient is high enough, this suggests that dust aerosols have a significant effect on the specific PS characteristic, regardless of the role of meteorological fields (Jiang et al., 2018; Han et al., 2022). For all sizes of PSs, all properties show a significant positive correlation with AOD, except for the surface

precipitation area of small- and medium- sized PSs and the convective precipitation percentage of large-sized PSs. This indicates again that PSs can develop more vigorously under dusty conditions. After removing the influence of meteorological factors, except for CAPE, most of the partial correlation coefficients can still pass the significance test, maintaining the same sign phase and changing by less than 20%. Even for CAPE, the relative change in the partial correlation coefficient to the total is less than 40%, which suggests that dust aerosols rather than meteorological conditions

contribute significantly to the observed variations in PS characteristics.

Interestingly, Zhu et al. (2023) also highlighted the important role of CAPE in the effects of dust on the vertical profiles of the precipitation growth rate in southeastern China, where stronger CAPE facilitates the vertical development of precipitation and leads to a decrease in precipitation top temperature. As mentioned earlier, CAPE values are higher in areas closer to land (Fig. S1a), which corresponds to dust-laden PSs (Fig. 2). This may be the main reason why CAPE has a

greater impact on the relationship between dust aerosols and PS characteristics than other meteorological factors do.





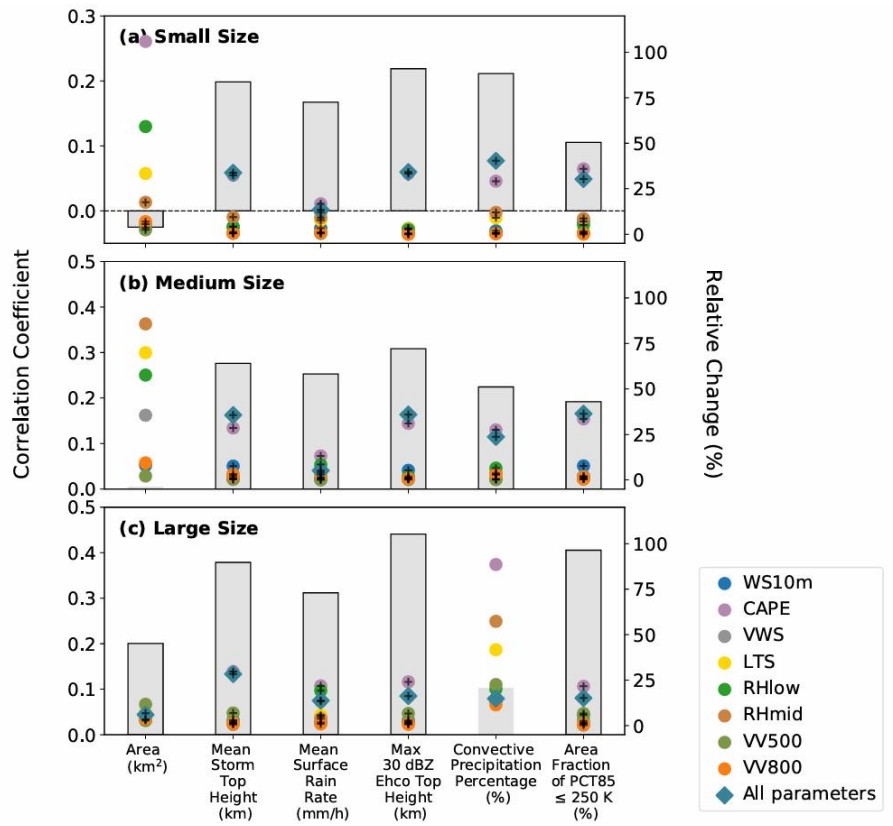

**Figure 7. Total correlations between AOD and PS properties (bars, left y-axis), and the relative changes in partial correlations with the effects of 8 meteorological parameters eliminated individually (colored circular markers) and totally (diamond-shaped markers), compared with total correlations (right y-axis) for (a) small-sized, (b) medium-sized and (c) large-sized PSs. The black border around grey data bar and the black cross on the marker indicate that the total correlation and the partial correlation are statistically significant at the 95% confidence level, individually.**

To clarify the influence of CAPE, PSs of each size were divided into 5 groups with similar sample numbers as CAPE increased (Table 5). Figure 8 shows the variations in the general characteristics of PSs of different sizes as a function of CAPE under clean and dusty environments, respectively. Clearly, regardless of dust loading or PS size, all the characteristics except for the surface precipitation area, increase with CAPE. This reflects the influence of thermodynamic conditions on PSs, as higher CAPE allows PSs to develop more vigorously. On the other hand, regardless of the CAPE group and PS size, it always shows that for dust-laden PSs, the area, mean storm top height, mean surface rain rate, maximum 30 dBZ echo top height, convective precipitation percentage and area fraction of PCT85 ≤ 250 K are higher compared to those for dust-free PSs, with most of these changes passing significance tests. This demonstrates that the enhancing effect of dust aerosols on PSs can still be observed with constrained CAPE conditions.



**Table 5. The CAPE ranges and sample numbers for PSs.**

| | Small-sized | | | Medium-sized | | | Large-sized | | |
| --- | --- | --- | --- | --- | --- | --- | --- | --- | --- |
| | Range (J/kg) | Clean | Dusty | Range (J/kg) | Clean | Dusty | Range (J/kg) | Clean | Dusty |
| CAPE1 | 0-270 | 2017 | 839 | 0-240 | 157 | 61 | 0-270 | 10 | 10 |
| CAPE2 | 270-410 | 1782 | 922 | 240-370 | 132 | 90 | 270-340 | 13 | 8 |
| CAPE3 | 410-570 | 1593 | 1069 | 370-490 | 127 | 92 | 340-440 | 15 | 4 |
| CAPE4 | 570-820 | 1220 | 1395 | 490-660 | 117 | 103 | 440-620 | 8 | 12 |
| CAPE5 | 820- | 621 | 2059 | 660- | 53 | 155 | 620- | 5 | 14 |

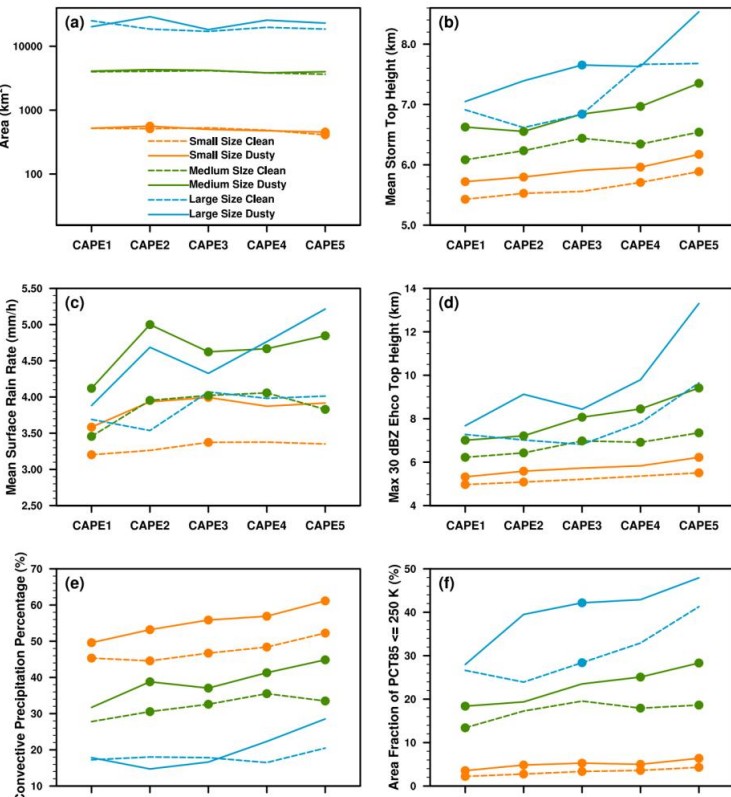

**Figure 8. Variations in PS properties, including (a) area (km²), (b) mean storm top height (km), (c) mean surface rain rate (mm/h), (d) maximum 30 dBZ echo top height (km), (e) convective precipitation percentage (%) and (f) area fraction of PCT85 ≤ 250 K (%), as a function of CAPE, for different sizes of PSs under clean and dusty conditions. The orange, green, and blue lines represent small-, medium- and large-sized PSs, respectively. The dashed and solid lines represent clean and dusty conditions, respectively. The circular marker indicates that the difference between clean and dusty conditions is statistically significant at the 95% confidence level using a Student's t-test.**



Also, the differences in the 3-D structure of PSs between clean and dusty conditions with various CAPE values are shown in Fig. 9 and Figs. S2-S3, taking small-sized PSs as an example. A similar conclusion is drawn that CAPE does not

350 substantially change the dust effect on the 3-D structure of PSs. For instance, in both convective and total precipitation, regardless of the CAPE level, dust-laden PSs have higher precipitation rates (Fig. S2), larger radar reflectivity areas (Fig. 9), and higher maximum radar reflectivity (Fig. S3) at all altitudes. This result further confirms the enhancement effect of dust aerosols on PSs. Interestingly, in both precipitation rate and radar reflectivity areas, the differences between pristine and dusty PSs increase as CAPE level rises. As shown in the partial correlation analysis, this indicates the modulation of

355 thermodynamic conditions on the dust effects, i.e., stronger thermodynamic forcing can amplify the dust effects, leading to stronger precipitation and convective intensity.

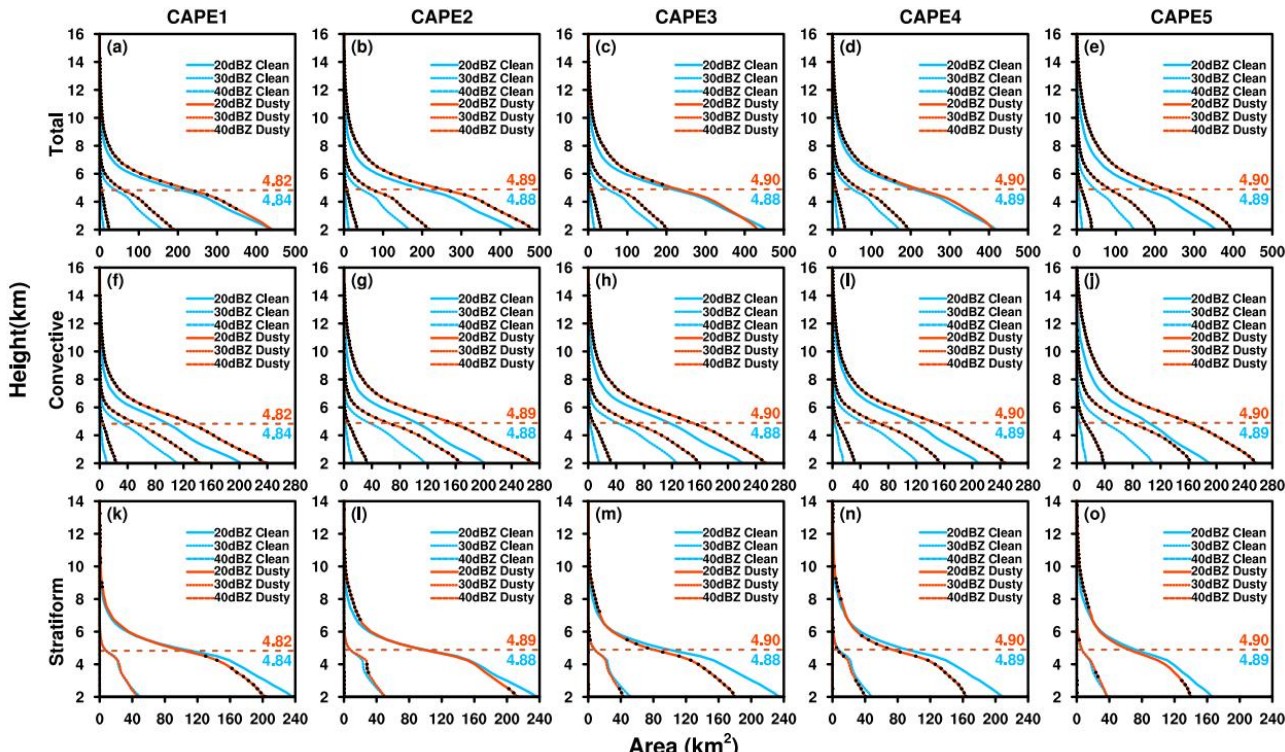

**Figure 9. The mean vertical profiles of 20 dBZ, 30 dBZ and 40 dBZ areas for total (upper row), convective (middle**
360 **row) and stratiform (bottom row) precipitation of small-sized PSs with different CAPE values under clean and dusty conditions. The black dot on the red line indicates that the difference between clean and dusty conditions is statistically significant at the 95% confidence level using a Student's t-test. The horizontal dashed lines show the freezing level with values labeled.**



## 4 Discussion

The scattering and absorption properties of dust would cause warming of the midtroposphere and cooling of the near surface, which increases low-level atmospheric stability and suppresses convection (Chaboureau and Martínez, 2018; Huang et al., 2006) (Fig. 10a). On the contrary, the microphysical effects of dust could increase the effective CCN and IN, leading to the formation of more cloud droplets and ice particles of smaller sizes. In turn, the heterogeneous freezing process is enhanced, and more numerous particles participate in depositional growth and riming, which increases latent heat release and invigorates the convective intensity of PSs thermodynamically (Gibbons et al., 2018) (Fig. 10c). Therefore, the enhancement effect of dust on PSs found in this study is more likely related to the aerosol microphysical effect rather than the radiative effects. Stronger updrafts of dust-affected PSs support the growth of more numerous particles and transport a greater number of large particles to higher altitudes in the convective core and into the adjoining stratiform regime. This results in a significant increase in the precipitation rate, maximum radar reflectivity and precipitation area at all heights in the convective regions, as well as those above the freezing level in the stratiform regions. Below the freezing level, however, due to the entrainment of ambient dry and warm air caused by the SAL, raindrops near the edge of PS are more prone to evaporation, resulting in a reduction in the stratiform precipitation area.

It should be mentioned that the current statistical results of PSs are somewhat different from previous case studies conducted over the tropical Atlantic Ocean (Li and Min, 2010; Dong et al., 2018), where the convective precipitation rate is lower at all heights in the presence of dust aerosols. One possible reason for this discrepancy is the difference in sample size, with previous studies using cases of less than a month, compared with more than 10 years of observations here. Additionally, this study focuses on completely observed PSs that have developed to at least 6 km, whereas some other studies have mainly performed rainfall profile statistics at the pixel level (Li and Min, 2010; Dong et al., 2018). Apart from differences in data sample size and research subjects, the discrepancy may also be related to differences in the background field, such as water vapor and thermodynamic conditions. In convective regions of PSs, whether dust acts as IN or CCN, it tends to reduce the particle size and increase the particle numbers. This enhances competition for water vapor during particle growth, and reduces the efficiency of collection processes, thereby weakening precipitation when the water vapor supply is limited. Gibbons et al. (2018) found that convective cloud formation is weakened when reducing water vapor content based on simulations. In this statistical study, however, the chosen PS samples are mainly located in the ITCZ region (Fig. 2) with abundant water vapor conditions, which are favorable for particle growth and the development of convective precipitation.



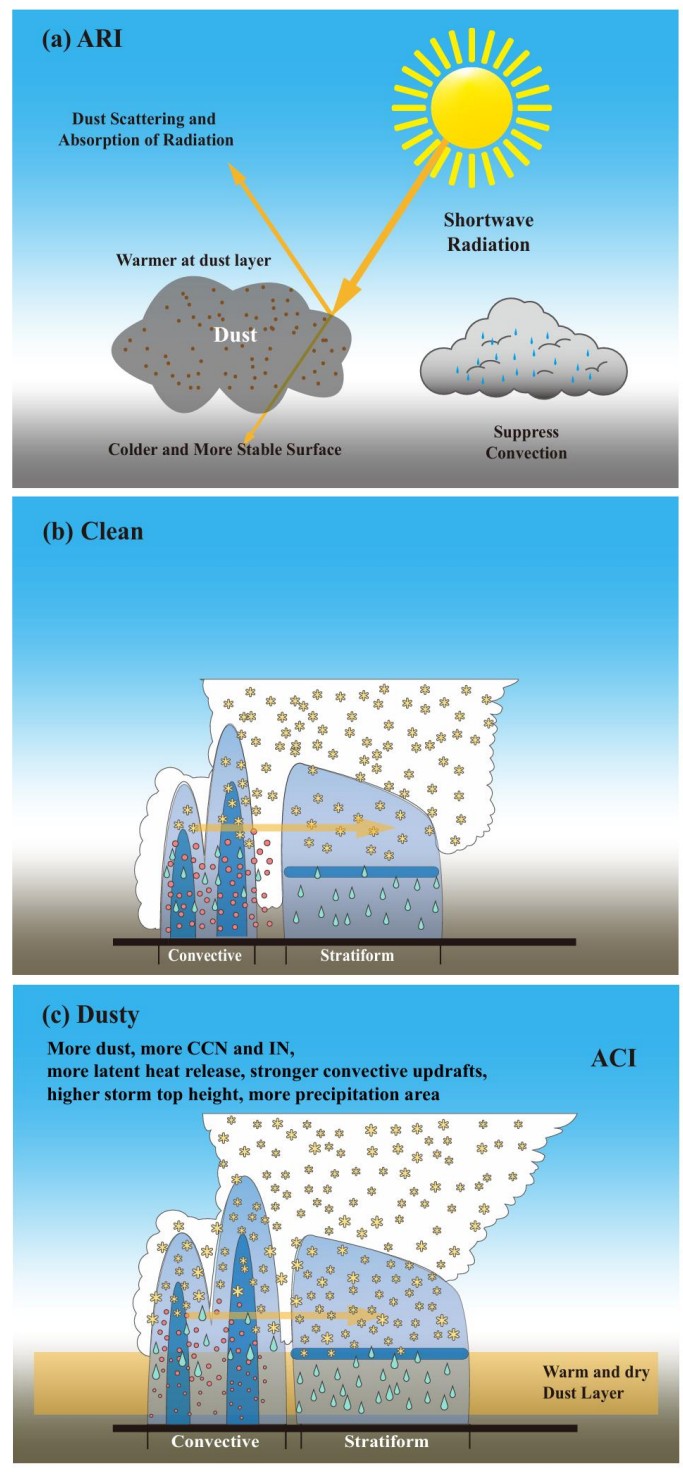

**Figure 10. Schematic plot of dust aerosol effects over the tropical Atlantic Ocean.**



## 5 Conclusions

Mineral dust is an important component of the Earth system. It can not only influence the global radiation budget by scattering and absorbing shortwave and longwave radiation, but also can alter cloud microphysical properties by acting as cloud condensation nuclei (CCN) and ice nuclei (IN) (Knippertz and Stuut, 2014; Ryder et al., 2019). To investigate dust effects on organized precipitation systems (PSs), a statistical comparison was conducted on the general characteristics and 3-D structure of PSs with different horizontal sizes under clean and dusty conditions over the tropical Atlantic Ocean during

spring from 2003 to 2014, using long-term observations from multiple satellite-borne sensors, as well as reanalysis data.

The results show that dust aerosols can invigorate the development of PSs, with significant increases in various general characteristics, including the mean storm top height, mean surface rain rate, maximum 30 dBZ echo top height, convective precipitation percentage and area fraction of PCT85 ⩽ 250 K. In terms of the 3-D structure of PSs, the precipitation rate, the precipitation area (20 dBZ, 30 dBZ, and 40 dBZ), and the maximum radar reflectivity with increased dust aerosols were also

significantly enhanced, except for stratiform precipitation in small- and medium-sized PSs below the freezing level due to the evaporation effect. In addition, the analysis of potential meteorological influences indicates that convective available potential energy (CAPE) is the primary confounding factor for dust aerosol effects, but after constraining the variations in CAPE, significant dust-related PS changes persist.

It should be noted that the precipitation data used in this study are from the precipitation radar (PR) onboard the Tropical

Rainfall Measuring Mission (TRMM) satellite. Although PR can offer a long-time span and a large sample size of PSs for statistical analysis, it cannot provide accurate precipitation particle size information, which limits our understanding of the microphysical effects of dust. Therefore, future work will utilize observations from the dual-frequency precipitation radar (DPR) on the Global Precipitation Measurement (GPM) satellite to obtain more detailed microphysical information within PSs. Also, the PR can only detect precipitation particles, not smaller cloud particles, so that the dust effects on cloud

properties and cloud cover are not included in our study. This merits further analysis using more comprehensive datasets in which cloud clusters and PSs can be well matched.

## Data availability

Precipitation data were obtained from the Tropical Rainfall Measuring Mission (TRMM) satellite product

(https://gpm.nasa.gov/missions/trmm). Aerosol optical depth data were obtained from the Moderate Resolution Imaging Spectroradiometer (MODIS) product (https://modis.gsfc.nasa.gov/data/). Hourly meteorological data were obtained from European Centre for Medium-Range Weather Forecasts ERA5 reanalysis (https://www.ecmwf.int/).



**Author contributions**

**JX:** Conceptualization, Methodology, Validation, Formal analysis, Software, Visualization, Writing - Original Draft. **YW:** Conceptualization, Methodology, Validation, Formal analysis, Writing - Original Draft, Project administration, Supervision, Funding acquisition. **RL:** Conceptualization, Methodology, Validation, Formal analysis, Writing - Original Draft. **BW:** Software. **XF, XM and ZM:** Writing-Original Draft.

**Competing interests**

The contact author has declared that none of the authors has any competing interests.

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
