# Peer review of "The impact of Sahara dust aerosols on the three-dimensional structure of precipitation systems of different sizes in spring"

_EGUsphere, 2025_

## Author Comment (AC3)

**Responses to Reviewers' Comments**

We are sincerely grateful to the editor and reviewers for their valuable time spent on reviewing our manuscript. The comments are constructive and valuable, and we have tried our best to address the issues raised by the reviewers in revised manuscript. Please find our item-by-item response (in bold) to the comments (in black) raised by the reviewers, and the paragraphs added in the revised manuscript (in red).

**Reviewer #1**

General Comments: The aim of this manuscript is to statistically understand the effect of dust aerosols on the three-dimensional structure of precipitation systems of different sizes. This is an interesting and valuable attempt, as it is not common to study aerosol effects based on a large number of observational samples with the whole precipitation system as the research unit. Also, the authors have carefully considered the influence of meteorological conditions and employed multiple approaches (e.g., partial correlation analysis, and CAPE constraint) for investigation. So I think this work is well-constructed and scientifically meaningful, hence can be accepted for publication after the minor issues are addressed.

Reply: We thank the reviewer for the valuable time and constructive comments, which have helped us to improve our manuscript. All comments have been addressed item by item.

**Major Comments:**

Q1: L177-180: The authors categorized PSs into three types: small-sized (< 2000 km), medium-sized (between 2000 km 2 and 10000 km 3), and large-sized (> 10000 km). Please clarify the reason for selecting these specific thresholds.

Reply: Thank you for raising this point. The area thresholds used in this study were determined based on previous studies on the characteristics of PSs. For instance, Liu et al. (2019) classified PSs with areas >2000 km 2as mesoscale convective systems (MCSs) in their analysis on the intensity, height, and size variations of PSs under El Niño—Southern Oscillation conditions in the tropics and subtropics. They also found that PSs exceeding 10,000 km 2contribute significantly to the annual mean rainfall (Fig. 7e in their paper). Similar thresholds have also been widely adopted in other studies (Zipser et al., 2008; Liu et al., 2017). The relevant references for the thresholds have been added in the manuscript.

Liu, C., Chen, B., and Mapes, B. E.: Relationships between Large Precipitating Systems and Atmospheric Factors at a Grid Scale, Journal of the Atmospheric Sciences, 74, 531-552,

10.1175/jas-d-16-0049.1, 2017.

Liu, N., Liu, C., and Lavigne, T.: The Variation of the Intensity, Height, and Size of Precipitation Systems with El Niño—Southern Oscillation in the Tropics and Subtropics, Journal of Climate, 32, 4281-4297, 10.1175/jcli-d-18-0766.1, 2019.

Zipser, E. J., Liu, C., Cecil, D. J., Nesbitt, S. W., and Sherwood, S.: A Cloud and Precipitation Feature Database from Nine Years of TRMM Observations, Journal of Applied Meteorology and Climatology, 47, 2712-2728, 10.1175/2008jamc1890.1, 2008.

Q2: Figure 3: Why do other characteristics of PSs significantly increase under dusty conditions, while the differences in PS areas between clean and dusty conditions are not apparent?

Reply: It can be seen that the precipitation areas are larger at most heights under the dusty conditions (Fig. 5). However, for stratiform precipitation, the areas below the freezing level decrease under high dust loading due to the evaporation effect. It should be noted that the precipitation area in Fig. 3 is calculated based on the number of raining pixels with PR near-surface rain rate greater than 0 mm/h, which is closer to the changes in the lower layers (i.e., below the freezing level) shown in Fig. 5. We have added the following clarification in the manuscript:

"It should be noted that the near-surface precipitation area shown in Fig. 3 primarily reflects the changes occurring in the lower layers (i.e., below the freezing level), as depicted in Fig. 5. Therefore, the differences in precipitation areas between clean and dusty conditions are not pronounced, although other characteristics of PSs increase significantly under dusty conditions."

Q3: Section 4: In the analysis of physical mechanisms, this manuscript mentioned both the CCN effect and the IN effect of dust, but failed to clearly distinguish between these two effects. A more explicit elaboration would be necessary and beneficial.

Reply: Thank you for this valuable comment. We agree that the CCN and IN effects of dust were not clearly distinguished in the original manuscript. In our study, both effects are expected to influence PS properties, but it is indeed challenging to clearly separate them based solely on observational data. To address this concern, we have added the following paragraph in the section of "Conclusions" of the revised manuscript:

"In our study, both the CCN and IN effects of dust contribute to changes of PS properties. Given the complexity of the microphysical processes within PSs, it is challenging to clearly separate these effects based solely on observational data. For future work, we propose conducting model simulation studies and statistical analyses of PSs that develop in warm clouds and comparing the results with those of PSs exceeding 6 km as considered in this study. This comparison would

help to understand the distinct roles of CCN and IN effect on precipitation structure."

**Minor Comments:**

Q1: L14: The term 'dimensions' in 'varying horizontal dimensions' may be misleading, as it refers to size rather than dimension here.

Reply: We have replaced 'dimensions' with 'sizes' in revised manuscript.

Q2: L21: I would recommend rephrasing the sentence 'significant dust-related Ps changes persist' to 'significant dust-induced changes in PS properties persist' for greater clarity and precision.

**Reply: Done!**

Q3: Figure 1: PS2 and PS4 have noticeably smaller areas compared with the other PSs. As stated in L109-110, only PSs larger than 80 km2 were selected. Therefore, these two PSs should be excluded from this part of the analysis to avoid potential misinterpretation.

Reply: We appreciate this helpful comment. To avoid potential misinterpretation, we have removed the identification of these two PSs in Fig. 1, and updated the table, and corresponding text accordingly.

"As an example, Figure 1 shows the spatial distributions of near-surface rain rate and AOD for the 4 PSs that occurred over Atlantic Ocean on 27 February 2008, and their general characteristics are given in Table 2. It can be seen that different PSs are highly variable in terms of the horizontal scale, as well as other properties and dust conditions. For example, PS2 and PS4 have similar horizontal sizes but exhibit noticeable differences in storm top heights and surface rain rates, implying differences in thermodynamic and dynamic conditions, or possibly aerosol effects. In particular, the higher maximum 30 dBZ echo top height and greater convective precipitation percentage in PS4 demonstrate stronger convective intensity, with a greater number of ice-phase particles. Moreover, it is interesting that PS1 and PS3 have different precipitation areas but similar other properties, such as maximum 30 dBZ echo top height, surface rain rate and convective precipitation percentage, which may be due to their locations in adjacent regions, resulting in relatively consistent background fields and dust aerosol concentrations."

Figure 1. Near-surface rain rates of several PSs idealized as ellipses over the tropical Atlantic Ocean on 27 February 2008 measured by PR (orbit number: 58594). The shaded map is the spatial distribution of MODIS AOD. The four PSs are numbered and represented by black elliptical outlines.

Table 2. The general characteristics of the 4 PSs.

|                                         | PS1     | PS2    | PS3    | PS4    |
|-----------------------------------------|---------|--------|--------|--------|
| Mean AOD                                | 0.73    | 0.75   | 0.75   | 0.77   |
| Near-surface Precipitation Area (km 3)  | 2849.00 | 529.10 | 752.95 | 427.35 |
| Mean Storm Top Height (km)              | 9.08    | 6.38   | 9.20   | 8.13   |
| Mean Surface Rain Rate (mm/h)           | 4.74    | 1.88   | 4.57   | 2.86   |
| Maximum 30 dBZ Echo Top Height (km)     | 11.25   | 4.50   | 11.25  | 7.75   |
| Convective Precipitation Percentage (%) | 58.57   | 38.46  | 54.05  | 42.86  |
| Area Fraction of PCT85 ≤ 250 K (%)      | 65.00   | 0.00   | 43.24  | 19.05  |

Q4: Figure 6: Why are the significance levels of the differences between maximum radar reflectivity profiles of PSs under clean and dusty conditions not marked, as was done in the other profile figures?

Reply: We have replaced Figure 6 with a new version below, which includes the significance test results.

Figure 6. Vertical profiles of the 10th, 50th, and 90th percentiles of PS maximum radar reflectivity for total, convective and stratiform precipitation of small-sized (upper row), medium-sized (middle row), and large-sized (bottom row) PSs under clean and dusty conditions. The horizontal dashed lines show the freezing level with values labeled. The black dot on the red line of 90th percentile profile indicates that the difference between clean and dusty conditions is statistically significant at the 95% confidence level using a Student's t-test.

Q5: Figure 10: Numerous shape markers are used to represent different hydrometeors in the cloud, but these markers are not clearly labeled. A legend explaining this should be added.

Reply: The legend have been added.

Figure 10. Schematic plot of dust aerosol effects over the tropical Atlantic Ocean.

---

## Author Comment (AC4)

**Responses to Reviewers' Comments**

We are sincerely grateful to the editor and reviewers for their valuable time spent on reviewing our manuscript. The comments are constructive and valuable, and we have tried our best to address the issues raised by the reviewers in revised manuscript. Please find our item-by-item response (in bold) to the comments (in black) raised by the reviewers, and the paragraphs added in the revised manuscript (in red).

**Reviewer #2**

General Comments: This manuscript investigates the impact of dust aerosols on the three-dimensional structure of precipitation systems of different sizes using a variety of observational data and reanalysis data. The authors employed a clustering method to group satellite precipitation radar profiles into organized precipitation systems, which is a novel and valuable approach. Studying precipitation from the system perspective provides deeper insights into aerosol—precipitation interactions and their coupling with environmental conditions. However, there remain a few minor issues that need further clarification and refinement, as outlined below.

Reply: We thank the reviewer for the encouragement and the valuable comments to improve our manuscript. All the comments have been addressed in the revised manuscript.

**Major Comments:**

Q1: In section 2.1, since MODIS cannot detect aerosols below clouds, the authors used a spatiotemporal interpolation and extrapolation method to estimate the dust concentration. It is noted that the spatial extent of the extrapolation varies with the size of PSs, and the size of PSs can reflect cloud coverage to some extent. However, this manuscript does not evaluate the impact of cloud coverage. I believe sensitivity experiments should be added.

Reply: We appreciate this insightful question. We agree that the cloud coverage may influence the estimation of dust concentration. To address this issue, we have included the following discussion about sensitivity experiments in the supplementary materials:

"We have conducted sensitivity tests using Modern-Era Retrospective Analysis for Research and Applications for version 2 (MERRA-2) data. Specifically, we artificially removed varying proportions of valid data to simulate different cloud cover conditions. For each precipitation system (PS), the averaged MERRA-2 AOD in the PS region was taken as the "true" AOD. Then, these AOD data were removed (white blocks in Fig. S1), and additional values were randomly removed from surrounding areas (gray blocks) to represent different cloud fractions. Our

interpolation algorithm was then applied to the AOD data under varying cloud cover conditions, and compared with the true values. Figures S2 and S3 summarize the results.

Across different missing data fractions, the interpolated AOD agrees well with the "true" AOD, with root mean square error (RMSE) remaining low and correlation coefficients exceeding 0.8. Although performance slightly decreases with increasing missing data (e.g., declining correlation and slightly higher RMSE), the overall impact remains minor. This result is likely because the frequent Saharan dust outbreaks in the study region, which persist for several days. Thus, even under high cloud cover condition, valid data from surrounding grids and adjacent days still provide sufficient information to estimate dust aerosol conditions of PSs.

Figure S1. Spatial distribution of AOD for a PS case under varying proportions of artificially removed data. White blocks denote removed AOD data in PS region, and grey blocks denote additional removed data with different proportions from surrounding area.

Figure S2. Scatter plots of interpolated AOD (y-axis) versus true AOD (x-axis) for different proportions of artificially removed data.

Figure S3. Variations in (a) mean bias, (b) mean error, (c) RMSE, (d) standard bias, (e) standard error, and (f) correlation coefficient of the interpolated AOD relative to the true values for PSs of different sizes, as a function of the proportion of missing data. "

Q2: In section 3.2,the authors analyzed the influence of meteorological conditions on dust effects and found that CAPE plays a significant role. The author did not explain why CAPE emerges as a more prominent factor compared to other dynamic and moisture conditions(e.g., vertical wind shear, relative humidity). Such an explanation is critical for understanding of the complex interactions between precipitation, aerosols, and meteorology.

Reply: Thank you for your valuable comment. CAPE measures the convective instability energy, and is one of the most representative parameters of atmospheric dynamic conditions. CAPE can directly influence the vertical development of precipitation. It is expected that stronger CAPE conditions typically correspond to more vigorous convective processes, leading to higher precipitation rates and deeper cloud structures (Zhu et al., 2023).

Zhu, H., Li, R., Yang, S., Zhao, C., Jiang, Z., and Huang, C.: The impacts of dust aerosol and convective available potential energy on precipitation vertical structure in southeastern China as seen from multisource observations, Atmos. Chem. Phys., 23, 2421-2437, doi:10.5194/acp-23-2421-2023, 2023.

Q3: In section 4, although the radiative effect of dust can cause warming of the midtroposphere and cooling of the near surface, thereby suppressing convection, the influence of dust radiative effects cannot be ruled out. This manuscript provides a rather limited introduction to dust radiative effects, which requires further elaboration.

Reply: We fully agree with the reviewer that the radiative effects of dust should be discussed more thoroughly. We have added the following paragraph in the section of "Discussion" of the revised manuscript:

"Although the microphysical effects are considered to play a dominant role, the possible contribution of radiative effects cannot be excluded. A few modeling studies (e.g., Cheng et al., 2019) have shown that dust radiative effects can delay convection initiation, allowing for energy accumulation, which may ultimately lead to more intense convective development once triggered. This highlights the complexity of dust radiative effects, thus it is difficult to quantify their impact using observational data. Future modeling studies will be needed to conduct sensitivity experiments to disentangle the contributions of dust radiative and microphysical effects.

Cheng, C.-T., Chen, J.-P., Tsai, I. C., Lee, H.-H., Matsui, T., Earl, K., Lin, Y.-C., Chen, S.-H., and Huang, C.-C.: Impacts of Dust–Radiation versus Dust–Cloud Interactions on the Development of a Modeled Mesoscale Convective System over North Africa, Monthly Weather Review, 147, 3301-3326, 10.1175/mwr-d-18-0459.1, 2019. "

Q4: The PSs selected for this study primarily consist of deep convective cloud systems that exceed 6 km in height. This selection criterion may affect the statistical results. Therefore, this should be highlighted in the conclusion section, so readers are aware of its potential influence on the findings.

Reply: We appreciate this suggestion. The selection criterion of PSs may influence the statistical results and should be clearly emphasized. We have added the following paragraph in the section of "Conclusions" of the revised manuscript:

"Our study primarily focuses on deep convective PSs with vertical development exceeding 6 km. Whether these conclusions can be extended to other types of PSs requires further investigation."

**Minor Comments:**

Q1: In line 38, 'with a temperature of between -5 and +2  $^{\circ}$ C' should be corrected to 'with temperatures between -5 and +2  $^{\circ}$ C'.

**Reply: Done!**

Q2: In Table 5, the sample sizes of large-sized PSs across different CAPE bins appear too small, which raises concerns about the statistical reliability of the results.

Reply: Thanks for pointing this out. We agree with the reviewer that the small sample sizes of large-sized PSs across different CAPE bins may affect the statistical reliability of the results. We have added the following paragraph in the revised manuscript when presenting Table 5:

"It is noted that the sample sizes for large-sized PSs across different CAPE bins are relatively small in Table 5. This is mainly due to the lower frequency of large-sized PSs compared to smaller ones, and the limitations of the PR swath, as only untruncated PSs were selected for analysis. While significance tests were applied to the data, the limited sample sizes may affect the statistical robustness of the results, which should be interpreted with caution."

Q3: In Figure 3, it would be more accurate to replace 'area' with 'near-surface precipitation area', since it is defined as the number of pixels with near-surface precipitation rates greater than 0 mm/h multiplied by the pixel area.

Reply: We have replaced 'area' with 'near-surface precipitation area' in Fig. 3, Fig. 7, and Fig. 8, and throughout the manuscript.

Figure 3. The averages of general characteristics of PSs of different sizes, including (a) near-surface precipitation area (km  $\frac{3}{2}$ , (b) mean storm top height (km), (c) mean surface rain rate (mm/h), (d) maximum 30 dBZ echo top height (km), (e) convective precipitation percentage (%) and (f) area fraction of PCT85  $\leq$  250 K (%) under clean and dusty conditions. The average values of these properties are labeled on the data bars. The black border around the data bar indicates that the difference between clean and dusty conditions is statistically significant at the 95% confidence level using a Student's t-test.

Figure 7. Total correlations between AOD and PS properties (bars, left y-axis), and the relative changes in partial correlations with the effects of 8 meteorological parameters eliminated individually (colored circular markers) and totally (diamond-shaped markers), compared with total correlations (right y-axis) for (a) small-sized, (b) medium-sized and (c) large-sized PSs. The black border around grey data bar and the black cross on the marker indicate that the total correlation and the partial correlation are statistically significant at the 95% confidence level, individually.

Figure 8. Variations in PS properties, including (a) near-surface precipitation area (km  $\frac{3}{2}$ , (b) mean storm top height (km), (c) mean surface rain rate (mm/h), (d) maximum 30 dBZ echo top height (km), (e) convective precipitation percentage (%) and (f) area fraction of PCT85  $\leq$  250 K (%), as a function of CAPE, for different sizes of PSs under clean and dusty conditions. The orange, green, and blue lines represent small-, medium- and large-sized PSs, respectively. The dashed and solid lines represent clean and dusty conditions, respectively. The circular marker indicates that the difference between clean and dusty conditions is statistically significant at the 95% confidence level using a Student's t-test.

Q4: The expression 'meteorology conditions' should be replaced with 'meteorological conditions' throughout the manuscript for grammatical accuracy and consistency.

**Reply: Done!**